# Stimulation of Potent Humoral and Cellular Immunity via Synthetic Dual-Antigen MVA-Based COVID-19 Vaccine COH04S1 in Cancer Patients Post Hematopoietic Cell Transplantation and Cellular Therapy

**DOI:** 10.3390/vaccines11091492

**Published:** 2023-09-15

**Authors:** Flavia Chiuppesi, Sandra Ortega-Francisco, Miguel-Angel Gutierrez, Jing Li, Minh Ly, Katelyn Faircloth, Jada Mack-Onyeike, Corinna La Rosa, Sandra Thomas, Qiao Zhou, Jennifer Drake, Cynthia Slape, Paolo Fernando, Wasima Rida, Teodora Kaltcheva, Alba Grifoni, Alessandro Sette, Angela Patterson, Shannon Dempsey, Brian Ball, Haris Ali, Amandeep Salhotra, Anthony Stein, Nitya Nathwani, Michael Rosenzweig, Liana Nikolaenko, Monzr M. Al Malki, Jana Dickter, Deepa D. Nanayakkara, Alfredo Puing, Stephen J. Forman, Randy A. Taplitz, John A. Zaia, Ryotaro Nakamura, Felix Wussow, Don J. Diamond, Sanjeet S. Dadwal

**Affiliations:** 1Department of Hematology and HCT, Hematologic Malignancies Research Institute, City of Hope National Medical Center, Duarte, CA 91010, USA; fchiuppesi@coh.org (F.C.); sortegafrancisc@coh.org (S.O.-F.); migugutierrez@coh.org (M.-A.G.); jinli@coh.org (J.L.); katelyn.faircloth@duke.edu (K.F.); jmackonyeike@coh.org (J.M.-O.); clarosa@coh.org (C.L.R.); sthomas@coh.org (S.T.); qzhou@coh.org (Q.Z.); tkaltcheva@coh.org (T.K.); apatterson@coh.org (A.P.); sdempsey@coh.org (S.D.); brball@coh.org (B.B.); harisali@coh.org (H.A.); asalhotra@coh.org (A.S.); astein@coh.org (A.S.); nnathwani@coh.org (N.N.); mrosenzweig@coh.org (M.R.); liananik@yahoo.com (L.N.); malmalki@coh.org (M.M.A.M.); jdickter@coh.org (J.D.); dnanayakkara@coh.org (D.D.N.); sforman@coh.org (S.J.F.); rnakamura@coh.org (R.N.); fwussow@coh.org (F.W.); 2Clinical Trials Office, City of Hope National Medical Center, Duarte, CA 91010, USA; jdrake@coh.org (J.D.); cslape@coh.org (C.S.); pfernando@coh.org (P.F.); 3Independent Researcher, Arlington, VA 22205, USA; wrida@aol.com; 4Division of Vaccine Discovery, La Jolla Institute of Allergy and Immunology, University of California San Diego, La Jolla, CA 92037, USA; agrifoni@lji.org (A.G.); alex@lji.org (A.S.); 5Division of Infectious Diseases, City of Hope National Medical Center, Duarte, CA 91010, USA; rtaplitz@coh.org (R.A.T.); sdadwal@coh.org (S.S.D.); 6Department of Medicine, City of Hope National Medical Center, Duarte, CA 91010, USA; 7Center for Gene Therapy, City of Hope National Medical Center, Duarte, CA 91010, USA; jzaia@coh.org

**Keywords:** SARS-CoV-2, modified vaccinia Ankara (MVA), spike, nucleocapsid, hematopoietic cell transplantation (HCT), immunosuppression, phase 2 clinical trial, humoral response, cellular response, vaccination, COVID-19

## Abstract

Hematopoietic cell transplantation (HCT) and chimeric antigen receptor (CAR)-T cell patients are immunocompromised, remain at high risk following SARS-CoV-2 infection, and are less likely than immunocompetent individuals to respond to vaccination. As part of the safety lead-in portion of a phase 2 clinical trial in patients post HCT/CAR-T for hematological malignancies (HM), we tested the immunogenicity of the synthetic modified vaccinia Ankara-based COVID-19 vaccine COH04S1 co-expressing spike (S) and nucleocapsid (N) antigens. Thirteen patients were vaccinated 3–12 months post HCT/CAR-T with two to four doses of COH04S1. SARS-CoV-2 antigen-specific humoral and cellular immune responses, including neutralizing antibodies to ancestral virus and variants of concern (VOC), were measured up to six months post vaccination and compared to immune responses in historical cohorts of naïve healthy volunteers (HV) vaccinated with COH04S1 and naïve healthcare workers (HCW) vaccinated with the FDA-approved mRNA vaccine Comirnaty^®^ (Pfizer, New York, NY, USA). After one or two COH04S1 vaccine doses, HCT/CAR-T recipients showed a significant increase in S- and N-specific binding antibody titers and neutralizing antibodies with potent activity against SARS-CoV-2 ancestral virus and VOC, including the highly immune evasive Omicron XBB.1.5 variant. Furthermore, vaccination with COH04S1 resulted in a significant increase in S- and N-specific T cells, predominantly CD4^+^ T lymphocytes. Elevated S- and N-specific immune responses continued to persist at six months post vaccination. Furthermore, both humoral and cellular immune responses in COH04S1-vaccinated HCT/CAR-T patients were superior or comparable to those measured in COH04S1-vaccinated HV or Comirnaty^®^-vaccinated HCW. These results demonstrate robust stimulation of SARS-CoV-2 S- and N-specific immune responses including cross-reactive neutralizing antibodies by COH04S1 in HM patients post HCT/CAR-T, supporting further testing of COH04S1 in immunocompromised populations.

## 1. Introduction

Hematopoietic cell transplantation (HCT) and chimeric antigen receptor (CAR)-T cell cancer patients are at increased risk of hospitalization and death following SARS-CoV-2 infection [1,2,3]. The use of COVID-19 vaccines in this population has resulted in better outcomes, but vaccine response rates remain below those measured in healthy volunteers (HV) [4,5,6,7,8]. This is due to suppressed immunity post cell therapy and is further worsened by transplantation-related morbidities [9].

Two doses of FDA-approved mRNA vaccines have been shown to be poorly immunogenic in HCT/CAR-T patients and often require subsequent doses to improve the response rate [4,10,11,12,13]. For this reason, current guidelines recommend three doses as a primary series followed by one or more booster doses with updated bivalent mRNA vaccines [14,15]. Major risk factors for low vaccine responses have been shown to include vaccination less than 12 months from transplant, low lymphocyte count, and use of immunosuppressive drugs [5,9,16,17].

We are conducting a phase 2 clinical trial to test the safety and immunogenicity of dual-antigen COVID-19 vaccine COH04S1, a fully synthetic modified vaccinia Ankara (sMVA) vector co-expressing SARS-CoV-2 Wuhan-Hu-1-based spike (S) and nucleocapsid (N) antigens [18,19]. In a phase 1 clinical trial in HV, COH04S1 has been shown to be safe and to induce robust antigen-specific humoral and cellular immune responses [20,21]. Additionally, COH04S1 demonstrated protection in small and large animal models against SARS-CoV-2 ancestral virus and several variants of concern (VOC), including Omicron subvariants [19,22,23].

In this study, we demonstrate robust stimulation of S- and N-specific humoral and cellular immune responses in HCT/CAR-T patients vaccinated with COH04S1 within one year post cell therapy. This includes neutralizing antibodies (NAb) with potent reactivity against SARS-CoV-2 ancestral virus and several variants of concern (VOC), including the Beta, Delta, Omicron BA.1, and the highly immune evasive Omicron XBB.1.5 variants. In addition, humoral and cellular responses elicited by COH04S1 in HCT/CAR-T patients surpass or are comparable to responses measured in HV vaccinated with COH04S1 as well as healthcare workers (HCW) vaccinated with the FDA-approved mRNA vaccine Comirnaty^®^ (BNT162b2, Pfizer, New York, NY, USA). These results support further clinical testing of COH04S1 in individuals with weakened immune systems.

## 2. Materials and Methods

### 2.1. Study Design

The randomized, multi-center, observer-blind study of COH04S1 versus Comirnaty^®^ in patients post HCT/CAR-T for hematological malignancies (HM) was approved by an external institutional review board (IRB) and is registered (NCT04977024). The open-label safety lead-in portion, conducted at City of Hope, enrolled between September 2021 and September 2022 and included 13 patients. After giving informed consent, patients were vaccinated with two to four doses of COH04S1 at 2.5 × 10^8^ plaque forming units (pfu) diluted in PBS/7.5% lactose. Immunological assessment was carried out at baseline and at days 28, 56, 120, and 180. For patients receiving additional vaccine doses at days 56 and 140, subsequent blood draws were taken at days 84, 140, and 180 (Appendix A). Immunological results for patients at all timepoints following a reported positive COVID-19 test (4/13 patients), or after boosting with FDA-approved SARS-CoV-2 vaccines (3/13 patients) were not included in the immunological analysis (Appendix A and Appendix A). Historical cohorts of naïve HV vaccinated with COH04S1 [20] (NCT04639466, 2.5 × 10^8^ pfu, two doses) and naïve HCW vaccinated with two doses of Comirnaty^®^ [21] (City of Hope IRB#20720) were used as comparator. HV and HCW were enrolled between December 2020 and May 2021.

### 2.2. Binding IgG

To measure S-, receptor binding domain (RBD)-, and N-specific binding IgG, a quantitative ELISA calibrated with the WHO international standard serum was used as described previously [21]. Briefly, 1 μg/mL S, RBD, and N proteins (Sino Biological 40589-V08B1, 40592-V08H, 40588-V08B) in PBS were used to coat 96-well plates. After blocking in sample buffer (0.5% casein/154 mM NaCl/10 mM Tris-HCl/0.1% Tween-20 [pH 7.6]/8% Normal goat serum), 6-fold serial dilutions of serum in sample buffer were added to the plates. Plates were incubated for 2 h at 37 °C, after which anti-human IgG HRP secondary antibody (1:3000, BioRad, Hercules, CA, USA, 204005) in sample buffer was added for 1 h. WHO International reference panel (NIBSC, 20/268) assigned values were used to create a standard curve. For each sample, the first absorbance value to fall within the standard curve range was used to calculate the IgG titer and expressed as BAU/mL.

### 2.3. Neutralizing Antibodies

SARS-CoV-2-specific NAb against ancestral and variant viruses were measured by microneutralization assay using SARS-CoV-2 pseudovirus (PsV) [21]. Lentiviral-based PsV were produced using the pALD system (Aldevron, Fargo, ND, USA) and Spike expression plasmids with C-terminal 19-amino acid deletion (custom order, cloned into pTwist-CMV-BetaGlobin vector, Twist Biosciences, South San Francisco, CA, USA). Serial dilutions of serum starting from 1:20 were prepared in triplicate, mixed with PsV and added to poly-L-lysin-coated 96-well plates. Plates were incubated overnight at 4 °C after which 10,000 HEK293T-ACE2 cells [24] were added to each well in the presence of 3 μg/mL polybrene and incubated at 37 °C for 48 h. Luminescence was quantified using SpectraMax L (Molecular Devices, San Jose, CA, USA) after adding Luciferase Assay Reagent (Promega E1483).

### 2.4. ELISPOT

IFNγ/IL-4 secretion was evaluated on thawed PBMCs using the human IFNγ/IL-4 FluoroSpot FLEX kit (Mabtech, Nacka Strand, Sweden, X-01A16B) following manufacturer instructions. Briefly, 150,000 cells/well in CTL-test serum-free media (Immunospot CTLT-010, Immunospot, Shaker Heights, OH, USA) were added to duplicate wells and stimulated with S, N, and membrane (M) peptide pools (15-mers, 11 aa overlap, >70% purity; GenScript and in-house synthesis). The S peptide library was divided into 4 sub-pools spanning the S1 and S2 domains. Each peptide pool (2 μg/mL) and αCD28 (0.1 μg/mL, Mabtech) was added to the cells, and plates were incubated for 48 h at 37 °C. Primary and secondary antibodies were added according to the manufacturer’s protocol. Fluorescent spot forming units (SFU) were acquired using CTL S6 Fluorocore (Immunospot).

### 2.5. Activation-Induced Marker (AIM) Assay

S- and N-specific T-cell phenotype analysis was carried out using an AIM assay [25]. Briefly, 10^6^ PBMCs were stimulated for 24 h with 1 µg/mL of either S-15mer megapool [25] or N peptide pool, in RPMI media with 5% human serum (BioIVT). DMSO (equimolar amount) and PHA (20 µg/mL) were used as negative and positive controls, respectively. Cells were stained at room temperature with live/dead, anti-CD19, anti-CD14, anti-CD3, anti-CD4, anti-CD8, anti-OX40, anti-CD137, anti-CD69, anti-CCR7, and anti-CD45RA (Appendix A). After washing, cells were resuspended in PBS and acquired on the Attune NxT cytometer (Thermo-Fisher, Waltham, MA, USA). Data were analyzed with the FlowJo X software (v10.8). Gating strategy is shown in Appendix A.

### 2.6. Statistics

GraphPad Prism (v9.4.1) was used to calculate statistical power. Student’s t distribution with N–1 degrees of freedom was applied on log10-transformed responses to compare baseline values to post-vaccination timepoints. The two-tailed Mann–Whitney test was used to compare the immunogenicity of COH04S1 and Comirnaty^®^.

## 3. Results

A phase 2 randomized clinical trial investigating the safety and immunogenicity of COH04S1 compared to the FDA-approved mRNA vaccine Comirnaty^®^ in allogeneic (allo) and autologous (auto) HCT recipients and CAR-T cell therapy patients is ongoing at City of Hope (NCT04977024). The FDA-required open-label safety lead-in segment was designed with six patients in each treatment group (auto-HCT, allo-HCT, and CAR-T in a 3 + 3 design). Six auto-HCT, six allo-HCT, and one CAR-T patient were enrolled and received two doses of COH04S1 on days 0 and 28. Due to changes in the CDC vaccination guidelines, a protocol amendment was subsequently submitted to allow for a third and fourth COH04S1 vaccine dose at days 56 and 140, respectively. Five out of thirteen patients received a third dose, and four of these received an additional fourth dose (Appendix A). Of the thirteen patients, nine (69%) received the first dose 3–6 months post therapy, and four (31%) received the first dose 6–12 months post therapy. Pre-treatment diagnosis was heterogeneous. Concomitant immunosuppressive or immunotherapeutic treatments were allowed, except for high dose corticosteroids (>0.5 mg/kg/day). Most patients reported receiving FDA-approved COVID-19 vaccines and/or monoclonal antibodies before HCT/CAR-T. Three patients were boosted with an FDA-approved COVID-19 vaccine after receiving COH04S1, and four COVID-19 asymptomatic/mild-symptomatic cases were reported during the trial. Those patients were subsequently removed from the immune analysis. Complete patient characteristics are presented in Table 1 and Appendix A.

Pre-existing S- and RBD-specific binding antibodies were measured at baseline in most subjects. Vaccination with one and two doses of COH04S1 resulted in a significant increase in S- and RBD-specific IgG (0.0004 < *p* < 0.013), and elevated S- and RBD-specific IgG responses were maintained for up to six months (Figure 1A and Appendix A, and Appendix A). N-specific IgG were low to absent at baseline and were significantly elevated after two COH04S1 vaccine doses (d56 *p* = 0.0159), after which they remained relatively stable up to day 180. Using a pseudoviral-based neutralization assay, elevated NAb responses against the Wuhan-Hu-1 strain, and the Beta and Delta variants were measured at baseline in most patients (NT50 GM 37, 31, and 43, respectively), while Omicron BA.1 and XBB.1.5 variant-specific NAb responses were low to absent in most patients (NT50 GM 10). A significant increase in SARS-CoV-2-specific NAb titers against all strains was observed after one dose (d28 0.008 < *p* < 0.0079), and these NAb titers further increased after the second dose (d56 0.0013 < *p* < 0.0023) (Figure 1B and Appendix A, and Appendix A). In patients who received two doses and those receiving additional COH04S1 vaccine doses, NAb titers remained elevated for at least six months post vaccination. Importantly, cross-NAb responses against the highly NAb-evasive XBB.1.5 Omicron subvariant were measured in most patients after one and two doses (7/13 and 8/12, respectively), and in all patients after three or more COH04S1 doses (Figure 1B and Appendix A).

S-specific IFNγ T cell responses at baseline were above the arbitrary threshold of positivity for most patients. In contrast, N- and M-specific T cell responses measured at baseline were low to absent in most patients. Vaccination with one or two doses of COH04S1 significantly increased S- and N-specific IFNγ T cell responses compared to baseline (0.0001 ≤ *p* < 0.0002), while it had no impact on M-specific T cell levels (Figure 1C and Appendix A, and Appendix A). Significantly elevated S- and N-specific IFNγ T cells were measured up to six months post vaccination (d180 *p* = 0.0149 and 0.0063. Figure 1C). Although S- and N-specific IL-4 T cells significantly increased after COH04S1 vaccination (0.0068 < *p* < 0.04), they remained at low levels throughout the study, indicating a substantial Th1-biased cellular immune response (Appendix A). Interestingly, after vaccination with COH04S1, the only CAR-T cell patient of the study mounted a robust, albeit temporary, T cell response to S and N antigens despite the limited increase in SARS-CoV-2-specific humoral responses and the evident lack of response to a subsequent booster vaccination with Spikevax^®^ (mRNA-1273, Moderna, Cambridge, MA, USA) (Appendix A). A similar increase in SARS-CoV-2-specific NAb, and S- and N-specific IFNγ T cell responses was measured in auto- and allo-transplant recipients, and in patients that underwent HCT/CAR-T 3 to 6 and 6 to 12 months prior to COH04S1 vaccination (Appendix A), indicative of an immune response to vaccination with COH04S1 that is independent from the type and time of transplant.

In a subset of samples, analysis of activation-induced markers (AIM) was performed with flow cytometry to assess broader antigen-specific T cell responses. Consistent with the measured IFNγ-specific T cell responses by ELISPOT (Figure 1C), a significant increase in S- and N-specific CD4^+^ T cells was observed after one and two COH04S1 vaccine doses (0.0001 ≤ *p* < 0.0034) (Figure 1D and Appendix A, and Appendix A). S- and N-specific CD8^+^ T cells significantly increased compared to baseline after two vaccine doses (*p* = 0.0015 and 0.0009). Further analysis of naïve/memory subtypes after two vaccine doses revealed a predominance of S- and N-specific CD4^+^ T cells with effector memory (>70%) and central memory (>20%) phenotype, while S- and N-specific CD8^+^ T cells were predominantly effector memory (>30% effector memory and >30% terminally differentiated effectors) (Appendix A).

To better estimate the magnitude of the vaccine-elicited immune responses, S- and N-specific responses elicited in COH04S1-vaccinated HCT/CAR-T patients were compared to those elicited in naïve HV vaccinated with COH04S1 [20] and to those elicited by Comirnaty^®^ in naïve HCW [21] (Appendix A). Baseline S-specific humoral responses measured prior to vaccination were significantly higher in COH04S1-vaccinated HCT/CAR-T patients than in COH04S1- or Comirnaty^®^-vaccinated naïve volunteers (0.0001 < *p* < 0.0157. Figure 2A). S-specific IgG titers remained significantly elevated in COH04S1-vaccinated HCT/CAR-T patients compared to COH04S1-vaccinated HV or Comirnaty^®^-vaccinated HCW at both one and six months post vaccination (0.0001 < *p* < 0.0033. Figure 2A). Baseline N-specific humoral responses were low to absent in all three vaccine cohorts and similar between COH04S1-vaccinated HCT/CAR-T patients and COH04S1- or Comirnaty-vaccinated naïve volunteers (Figure 2B). Vaccination with COH04S1 induced comparable N-specific IgG titers in HCT/CAR-T patients and HV, and resulted in significantly elevated N-specific IgG titers compared to those measured in Comirnaty^®^-vaccinated HCW (0.0001 < *p* < 0.0008. Figure 2B).

COH04S1-vaccinated HCT/CAR-T patients showed elevated Wuhan-specific NAb responses at baseline compared to COH04S1- or Comirnaty^®^-vaccinated naïve volunteers, while NAb titers against Omicron XBB.1.5 at baseline were below the limit of detection in all subjects in the three vaccine cohorts with the exception of one HCT patient (Figure 2C,D). After two vaccine doses, at one and at six months post vaccination, NAb titers measured in COH04S1-vaccinated patients were significantly higher than those measured in COH04S1-vaccinated HV or Comirnaty^®^-vaccinated HCW for all tested SARS-CoV-2 variants, including Omicron XBB.1.5 (0.0001 ≤ *p* < 0.0331. Figure 2C,D and Appendix A). Strikingly, while XBB.1.5-specific NAb titers were below the detection limit in most COH04S1- and Comirnaty^®^-vaccinated HV and HCW, COH04S1 vaccination of HCT/CAR-T patients resulted in Omicron XBB.1.5 median NT50 titers above 10^2^. At peak response, SARS-CoV-2-specific NAb titers against all variants, except XBB.1.5, were consistently higher in Comirnaty^®^-vaccinated HCW than in COH04S1-vaccinated HV. In contrast, at six months post vaccination, NAb titers against all SARS-CoV-2 variants were similar between Comirnaty^®^-vaccinated HCW and COH04S1-vaccinated HV, albeit significantly lower than in COH04S1-vaccinated HCT/CAR-T patients (Appendix A).

Similar to the observed S-specific antibody and NAb responses, baseline S-specific T cells were significantly higher in COH04S1 HCT/CAR-T patients than in healthy adults vaccinated with COH04S1 or Comirnaty^®^, and they remained significantly higher after two vaccine doses, at both one and six months post vaccination (0.0001 < *p* < 0.0414. Figure 2E). No difference in baseline N-specific T cells was measured across cohorts, with most subjects showing IFNγ N-specific T cell levels below the threshold of positivity (Figure 2F). COH04S1 vaccination in both HCT/CAR-T patients and HV resulted in significantly increased N-specific T cells compared to Comirnaty^®^-vaccinated HCW at both one month and six months post vaccination (0.0001 ≤ *p* < 0.0012), with overall highest N-specific responses measured in COH04S1-vaccinated HCT/CAR-T patients (Figure 2F). M-specific T cells measured in both COH04S1 and Comirnaty^®^ vaccinees were uniformly very low throughout the time course (Appendix A).

## 4. Discussion

Numerous studies have reported on the poor response rate and durability of the immune response to mRNA vaccination in HCT/CAR-T patients, which is further complicated in patients vaccinated less than one year post therapy [5,26]. Here we demonstrate that a heterogeneous group of HCT/CAR-T patients vaccinated with two or more doses of dual-antigen sMVA-vectored COH04S1 vaccine less than one year post treatment can mount robust humoral and cellular responses to S and N antigens, including NAb, with potent cross-reactivity against the SARS-CoV-2 ancestral virus and several VOC. In addition, NAb and T cell responses elicited in COH04S1-vaccinated HCT/CAR-T patients significantly exceed those measured in a cohort of COH04S1-vaccinated HV or HCW vaccinated with the FDA-approved mRNA vaccine Comirnaty^®^.

The dual antigen design differentiates COH04S1 from FDA-approved COVID-19 vaccines that exclusively utilize S as an immunogen. While S-elicited NAb are considered the principal immune correlate of protection, S and in particular its RBD are known as mutational hotspots allowing SARS-CoV-2 to evolve into new variants with the ability to escape protective NAb responses [27]. In contrast, the N protein is considered less susceptible than the S protein to evasion by humoral and cellular immune responses [27]. N-specific antibodies have been shown to contribute to NK cell activation [28] and to enhance control of SARS-CoV-2 through NK-mediated antibody-dependent cellular cytotoxicity (ADCC) [29,30]. Importantly, there is strong evidence in animal and human studies that N-specific T cells play a major role in protection from severe disease [31,32,33]. Therefore, use of a dual antigen vaccine based on S and N should be encouraged, especially in immunocompromised individuals with limited ability to mount robust humoral and cellular responses through vaccination.

While the emergence of Omicron may have resulted in a more favorable outcome in patients undergoing HCT/CAR-T [34], the more recent surges of further evolved Omicron subvariants BQ.1.1 and XBB have de facto eliminated the possibility of benefiting from monoclonal antibody treatments such as Evusheld^®^ (tixagevimab/cilgavimab), which were extensively used in this patient population as a complement to vaccines and antivirals. Additionally, compared to previous variants, the appearance of Omicron has resulted in significantly increased breakthrough infections in boosted HCT/CAR-T patients [35]. There were four breakthrough cases of COVID-19 in our cohort (two in allo-HCT and two in auto-HCT patients), occurring during a time spanning consecutive Omicron waves, and it is encouraging that all these cases were asymptomatic/mild-symptomatic and did not require hospitalization.

While the auto-HCT patients had documented pre-transplant vaccination with FDA-approved vaccines, donor vaccination record for most allo-HCT patients was unavailable. However, it is likely that the majority of allo-HCT recipients were transplanted with a non-naïve donor graft, given that these transplants took place at a time when vaccination rates in the US and Europe reached up to 78% [36]. Multiple studies have shown that donor natural or vaccine-induced immunity can be transferred to the recipient and boosted via vaccination [37,38,39]. How COH04S1 compares to Comirnaty^®^ in stimulating SARS-CoV-2-specific immunity in this heavily treated, likely non-naïve population is being evaluated in the randomized blinded portion of the trial.

Vaccine-elicited preexisting immunity to S in the graft likely contributed to the stimulation of the increased S-specific humoral and cellular immune responses observed in COH04S1-vaccinated HCT, especially when compared to healthy naïve adults vaccinated with COH04S1 or Comirnaty^®^. This indicates that COH04S1 can effectively restimulate donor-derived S-specific memory B and T cells most likely elicited through mRNA vaccination. In contrast, N-specific immunity at baseline was low to absent in most patients, suggesting that N-specific responses occurred most likely de novo and were solely due to post-transplant response to COH04S1 vaccination. Consistent with other studies on HCT patients, we observed more than 90% of the circulating specific CD4^+^ T cells displaying a memory phenotype after two COH04S1 vaccine doses [40]. This is important considering the role of vaccination-elicited memory T cell responses for protection against emerging SARS-CoV-2 VOC [41].

Most patients received the COH04S1 vaccine 3–6 months post therapy, at a time when the T cell compartment is not fully reconstituted [42] and a T cell response to SARS-CoV-2 vaccination is often not mounted [43]. The SARS-CoV-2-specific T cell responses in these vaccinated patients using the sMVA-based COH04S1 vaccine is consistent with our previous observations of elevated CMV-specific cellular responses in HCT recipients vaccinated post HCT with an MVA-based Triplex cytomegalovirus (CMV) vaccine [44]. These prior findings and our current observations with COH04S1 indicate that MVA-based vaccines are highly immunogenic in this patient population at early timepoints post transplant. These observations are noteworthy, as the FDA has chosen the configuration of the updated mRNA vaccine for the fall 2023 rollout. The fact that COH04S1 provides strong recognition of Omicron subvariants without updating, and the knowledge that it causes potent stimulation of immunity that exceeds what we documented in the HV cohort, portends continued efficacy as Omicron evolves. Since the T cell response to S is less susceptible to antibody resistance mutations, these patients might be equally or better served by receiving a booster of COH04S1 rather than updated mRNA boosters.

## 5. Conclusions

Despite the small and heterogeneous population evaluated in this study, the remarkably robust S- and N-specific humoral and cellular responses observed in COH04S1-vaccinated HCT/CAR-T patients are encouraging. These observations warrant further studies with COH04S1 in other immunocompromised patient populations that are known to respond poorly to vaccination with approved mRNA vaccines.

## Figures and Tables

**Figure 1 vaccines-11-01492-f001:**
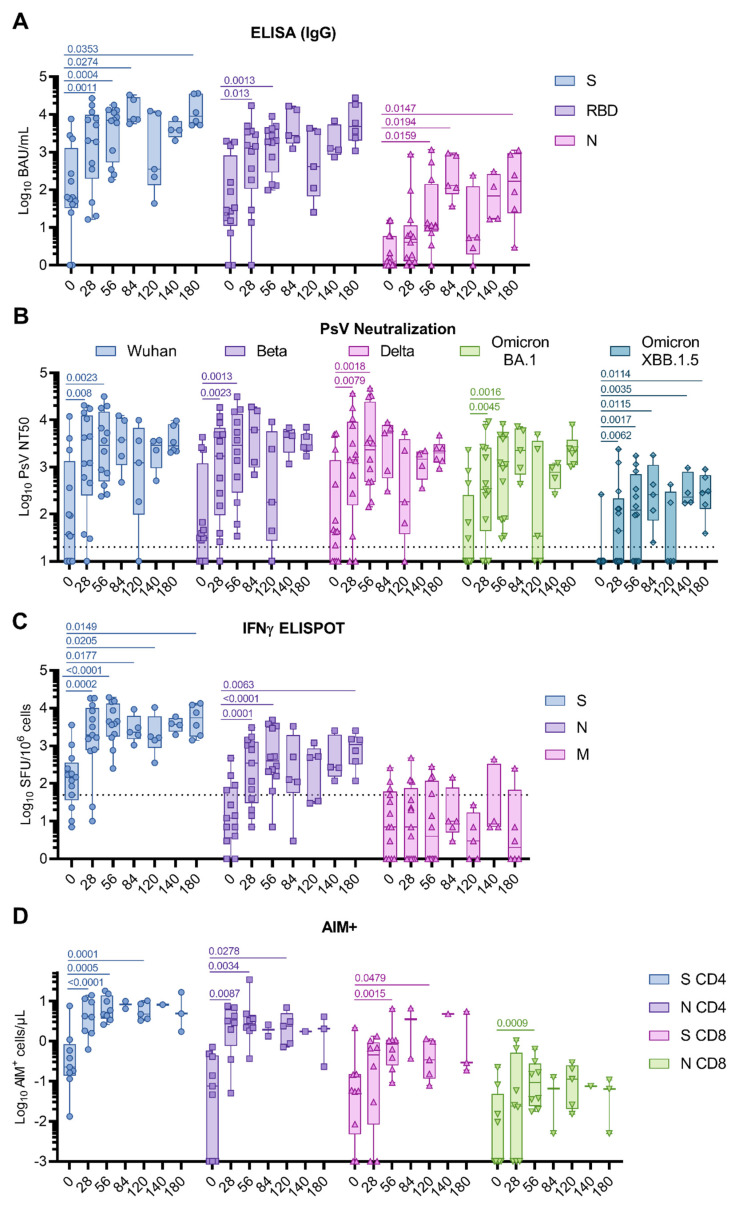
COH04S1-elicited humoral and T cell immunity in cancer patients post hematopoietic cell transplantation and cellular therapy. Patients (*n* = 13) were vaccinated two to four times with COH04S1, and blood samples were evaluated for SARS-CoV-2-specific humoral and T cell immunity at the indicated timepoints. (**A**) Binding antibodies. Binding antibody titers to spike (S, circles), receptor-binding domain (RBD, squares), and nucleocapsid (N, triangles) antigens were measured with quantitative ELISA. (**B**) NAb responses. 50% neutralizing antibody titers (NT50) against ancestral SARS-CoV-2 (Wuhan-Hu-1, circles), Beta (squares), Delta (up-pointing triangles), and Omicron BA.1 (down-pointing triangles) and XBB.1.5 (diamonds) variants were measured using a pseudovirus (PsV) assay. Dotted lines represent the lower limit of detection. (**C**) IFNγ T cells. IFNγ T cells were quantified via ELISPOT following stimulation of PBMCs with S (circles), N (squares), and membrane (M, triangles) peptide libraries. Shown are the IFNγ spot forming units (SFU) measured in 10^6^ PBMCs. Dotted line represents the arbitrary threshold for a positive response (50 SFU/10^6^ PBMCs). (**D**) Activation-Induced Markers (AIM^+^) T cells. CD4^+^ and CD8^+^ AIM^+^ T cells per μL of blood were quantified via cytofluorimetry in PBMCs stimulated with S (CD4^+^ and CD8^+^ AIM^+^ T cells indicated as circles and up-pointing triangles, respectively) and N (CD4^+^ and CD8^+^ AIM^+^ T cells indicated as squares and down-pointing triangles, respectively) peptide libraries. Data are presented as box plots extending from 25th to 75th percentile, with lines indicating medians, and whiskers going from minimum to maximum values. Student’s *t* test on log10-transformed data was applied to compare baseline to post-vaccine geometric mean fold rise (GMFR) values. *p* values ≤ 0.05 are indicated. Where not indicated, *p* > 0.05.

**Figure 2 vaccines-11-01492-f002:**
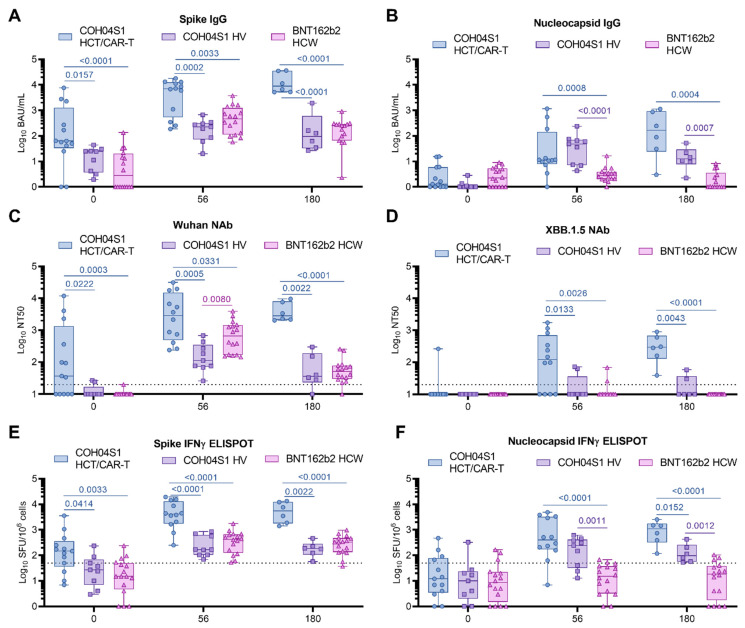
Immunogenicity of COH04S1 in HCT/CAR-T patients and COH04S1 or Comirnaty^®^ in healthy adults. SARS-CoV-2 binding antibodies (IgG) to SARS-CoV-2 spike (S, (**A**)), nucleocapsid (N, (**B**)), neutralizing antibody titers to SARS-CoV-2 Wuhan-Hu-1 ancestral strain (**C**), and XBB.1.5 variant (**D**), and IFNγ T cells specific for SARS-CoV-2 S (**E**) and N (**F**) were measured in samples of COH04S1-vaccinated HCT/CAR-T patients (circles) at baseline (*n* = 13), day 56 (*n* = 12), and 180 (*n* = 6) post vaccination, and compared to responses measured in healthy volunteers (HV, squares, d0-d56 *n* = 9, d180 *n* = 6) vaccinated with COH04S1 or healthcare workers (HCW, triangles, *n* = 17) vaccinated with the FDA-approved mRNA vaccine Comirnaty^®^ at the same timepoints. Dotted lines in (**C**,**D**) represent the lower limit of detection. Dotted lines in (**E**,**F**) represent the arbitrary threshold of positivity (50 spots/10^6^ cells). At the day 180 timepoint, 4/6 HCT/CAR-T patients had received three/four COH04S1 doses. Data are presented as box plots extending from 25th to 75th percentile, with lines indicating medians, and whiskers going from minimum to maximum values. Values were compared using the two-tailed Mann–Whitney test. *p* values ≤ 0.05 are indicated. Where not indicated, *p* > 0.05.

**Table 1 vaccines-11-01492-t001:** Total and grouped patient characteristics.

	COH04S1, HCT/CAR-T	COH04S1, Allo-HCT	COH04S1, Auto-HCT	COH04S1, CAR-T
Number of individuals, *n*	13	6	6	1
Age, median (Q1–Q3)	52 (46–66)	52 (43–62)	50 (46–68)	73 (/)
Female sex, *n* (%)	8 (62)	3 (50)	5 (83)	0 (0)
Race:				
White, *n* (%)	10 (77)	4 (67)	5 (83)	1 (100)
Black/African American, *n* (%)	1 (8)	/	1 (17)	/
Asian, *n* (%)	1 (8)	1 (17)	/	/
Other/Declined, *n* (%)	1 (8)	1 (17)	/	/
Hispanic/Latino ethnicity, *n* (%)	3 (23)	1 (17)	2 (33)	/
Baseline WBC/μL, median (Q1–Q3)	3600 (2800–5000)	3550 (2850–4800)	4400 (3075–6775)	2300 (/)
Baseline CD4^+^/CD8^+^ T-cell ratio, median (Q1–Q3)	1.55 (0.83–3.77)	1.79 (0.99–3.63)	2.14 (0.86–4.66)	0.37 (/)
Days post cellular therapy, median (Q1–Q3)	149 (121–217)	175 (141–207)	127 (105–171)	247 (/)

Q = Quartile.

## Data Availability

The authors confirm that the data supporting the findings of this study are available within the article and/or its Appendix A.

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
