# Peer review of "Stimulation of Potent Humoral and Cellular Immunity via Synthetic Dual-Antigen MVA-Based COVID-19 Vaccine COH04S1 in Cancer Patients Post Hematopoietic Cell Transplantation and Cellular Therapy"

_vaccines, 2023, doi:10.3390/vaccines11091492_

Round 1

Reviewer 1 Report

The manuscript demonstrates that the novel vaccine COH04S1 which carries two antigens and elicits strong humoral and T-cell responses in patients post-hematopoietic cell transplantation and cellular therapy.  This topic is exceedingly interesting to the readership since past studies have demonstrated poor vaccine responses in this patient cohort.  These findings suggest that use of the dual antigen vaccines offers a solution to this vulnerable population's susceptibility to SARS-CoV-2 infections. 

As the authors are aware, the primary limitation of the study is the low number of patients enrolled.  Despite this, the study findings potentially have broad and exciting implications for patients who have received stem cell transplantation. Once this study is published, it would allow for more robust clinical trials to be performed.  Of particular interest would be separation of allo-transplanted patients from patients who received autotransplants and cellular therapy to see if there are variations in responses between each cohort.  This would be of particular interest since prior studies have demonstrated that patients with myeloid neoplasms have better vaccine responses following transplantation compared to solid organ transplanted patients and that there may be some variations even in those patients who have had lymphoma therapy versus myeloid neoplasm or myeloma therapies.

Another arm that would be of interest in the future would be non-naive healthy volunteers. While it is notable that the transplanted patients had more robust humoral responses compared to naive HV, it is of interest to see how these responses compare to non-naive HV. 

The manuscript is clear, well-written and addresses most concerns within the scope of this well-designed study. 

Reviewer 2 Report

Dear Authors,

This article entitled "Stimulation of potent humoral and cellular immunity by synthetic dual-antigen MVA-based COVID-19 vaccine COH04S1 in cancer patients post hematopoietic cell transplantation and cellular therapy" is a very interesting orginal paper. Methodology and data analysis seem to me to be very appropriate and the results certainly interesting and relevant to the scientific community.  However, I have some comments to make to the authors.  Question 1: The introduction is a bit poor in information on the state of the art and in bibliographic references. I would suggest to the authors to increase its size making it more complete.

Question 2: In the results section, the authors include several comments trying to explain the results, even though such comments help to understand the outcome, the integration of these comments along with the material and method information in the results section makes it very complicated and redundant. I would suggest to the author to focus the commenting on the results only in the discussion section.

Question 3: In the discussion section, there are many long sentences that make the reader lost the track. I would suggest the author to plan an outline that would organize and order their comments on the results in a more simple and understandable way.

Except for these considerations I find this article of great interest and congratulate the authors. 

This is a novel approach and a well designed study with very interesting outcomes. 

My Best Regards.
